# The Effect of Replacing Genetically Modified Soybean Meal with 00-Rapeseed Meal, Faba Bean and Yellow Lupine in Grower-Finisher Diets on Nutrient Digestibility, Nitrogen Retention, Selected Blood Biochemical Parameters and Fattening Performance of Pigs

**DOI:** 10.3390/ani11040960

**Published:** 2021-03-30

**Authors:** Wiesław Sobotka, Elwira Fiedorowicz-Szatkowska

**Affiliations:** Department of Animal Nutrition and Feed Science, University of Warmia and Mazury in Olsztyn, Oczapowskiego 5, 10-719 Olsztyn, Poland; elwira.fiedorowicz@gmail.com

**Keywords:** growing-fattening pigs, alternative vegetable protein sources, nutrient digestibility, nitrogen retention, blood biochemical parameters, fattening performance, carcass quality

## Abstract

**Simple Summary:**

The aim of this study was to determine the effect of partial and total replacement of protein from genetically modified soybean meal (GM-SBM) with protein from 00-rapeseed meal (00-RSM), alone or in combination with protein from low-tannin faba bean seeds (FB) or low-alkaloid yellow lupine seeds (YL) in grower-finisher diets on nutrient digestibility, nitrogen retention, and utilization, selected blood biochemical parameters and fattening performance of pigs. During two-phase fattening, hybrid Danbred growing-finishing pigs were fed grower diets where 50% of GM-SBM protein was replaced with 00-RSM protein, 00-RSM and FB protein or 00-RSM and YL protein, and finisher diets where 100% of GM-SBM protein was totally replaced with 00-RSM protein, and with 50% 00-RSM and FB protein or YL protein. It was found that GM-SBM protein can be partially (50% in grower diets) and totally (100% in finisher diets) replaced with 00-RSM protein (6%) combined with protein from low-tannin FB seeds (10%/12%) or low-alkaloid YL seeds (6%/7%) in pig diets. The evaluated diets contributed to high nutrient digestibility and N retention and improved fattening performance without compromising the health status of pigs.

**Abstract:**

The aim of this study was to determine the effect of partial and total replacement of protein from genetically modified soybean meal (GM-SBM) with protein from 00-rapeseed meal (00-RSM), alone or in combination with protein from low-tannin faba bean (*Vicia faba* L.) seeds (FB) or low-alkaloid yellow lupine (*Lupinus luteus * L.) seeds (YL) in grower-finisher diets on nutrient digestibility, nitrogen retention and utilization, selected blood biochemical parameters, fattening performance of pigs and carcass quality traits. Two digestibility-balance trials and one feeding trial were performed during two-phase fattening on male hybrid Danbred growing-finishing pigs were divided into four groups. The pigs were fed grower diets where 50% of GM-SBM protein (diet S-c) was replaced with 00-RSM protein (diet R), 00-RSM and FB protein (diet R + FB) or 00-RSM and YL protein (diet R + YL), and finisher diets where 100% of GM-SBM protein (diet S-c) was totally replaced with 00-RSM protein (diet R), and with 00-RSM and FB protein (diet R + FB) or YL protein (diet R + YL) in 50%. It was found that partial (50% in grower diets) and total (100% in finisher diets) replacement of GM-SBM protein with 00-RSM protein combined with FB or YL protein had no adverse effect on nutrient and energy digestibility, N balance, serum of blood carbohydrate and protein metabolism or the biochemical parameters of liver and kidney function. Protein from 00-RSM (6%) and FB seeds (10%/12%) contributed to high daily gains and high feed conversion efficiency. Protein from 00-RSM (6%) and YL seeds (6%/7%) in grower-finisher diets led to a further improvement in fattening performance. The analyzed vegetable protein sources had no negative influence on carcass quality. The results of the present study indicate that 00-RSM protein combined with protein from low-tannin FB or low-alkaloid YL seeds can be valuable high-protein feed ingredients alternative to GM-SBM in pig nutrition.

## 1. Introduction

Modern pig fattening is based on complete diets with the nutritional value corresponding to the growth rate and protein deposition potential of animals. Pigs have high protein requirements; therefore, cereal-based diets must also contain high-protein components such as meals, including imported genetically modified soybean meal (GM-SBM) [1,2]. Recent years have witnessed considerable breeding progress in the genetic improvement of legumes. The seeds of new legume varieties are characterized by higher and more stable yields and a lower content of antinutritional factors [3]. Therefore, they can be fed to pigs to make up for the negative trade balance on high-protein ingredients used in monogastric nutrition.

There has been an ongoing debate in Poland and in other EU countries over eliminating GM crops, in particular GM-SBM, from pig feedstuffs [4,5,6,7]. Due to the social pressure exerted on the Polish government’s policies, restrictions on the production and use of genetically modified organisms (GMOs) and trade in GMOs may be introduced on 1 January 2023 in Poland, including a ban on GM components in animal feeds [8]. Moreover, GMOs are prohibited in organic animal production, which is becoming increasingly popular. High fluctuations in the prices and supply of GM-SBM and the need to provide affordable protein for animal feeds have triggered a search for alternative, cheaper local sources of vegetable protein [4,9,10,11]. In Poland, 00-rapeseed meal (00-RSM), faba bean (*Vicia faba* L.) seeds of new low-tannin varieties (FB), and yellow lupine (*Lupinus luteus* L.) seeds of new low-alkaloid varieties (YL) can be valuable protein sources in pig diets.

The results of previous experiments, though inconclusive, show that legume seeds should not be the sole protein source in pig diets [12,13]. Protein from FB and YL seeds has high lysine content and low concentrations of methionine and tryptophan. The ratio of lysine to methionine+cystine in these feedstuffs is 1:0.3–0.6 [14], whereas the ratio of the above amino acids in grower pig diets should be 1:0.65–0.70 [14]. Therefore, grain legumes cannot be the only source of supplemental protein in cereal-based pig diets. Protein from 00-RSM is characterized by high concentrations of methionine+cystine (2.1 and 2.5 g/16 g N, respectively) and a ratio of lysine to methionine+cystine of 1:0.83. It also has higher tryptophan content (1.3 g/16 g/N) than FB and YL protein. Therefore, 00-RSM protein can supplement FB and YL protein in diets for growing-finishing pigs.

The aim of this study was to determine the effect of partial and total replacement of GM-SBM protein with 00-RSM protein combined with protein from low-tannin FB seeds or low-alkaloid YL seeds in grower-finisher diets on nutrient digestibility, nitrogen retention and utilization, selected blood biochemical parameters, fattening performance of pigs and carcass quality traits.

## 2. Materials and Methods

The animal protocol and the number of animals used in this study were consistent with regulations of the Local Institutional Animal Care and Use Committee (23/2013 Olsztyn, Poland), and the study was carried out in accordance with EU Directive 2010/63/EU on the protection of animals used for scientific purposes [15].

### 2.1. High-Protein Vegetable Feed Ingredients

The chemical and amino acid composition of protein and the content of antinutritional factors in GM-SBM, 00-RSM, seeds of FB cv. Albus, and YL cv. Taper are presented in Table 1.

Chemical analysis of protein quality in the analyzed high-protein vegetable feedstuffs was performed (Table 1), and the results were expressed as the essential amino acid index (EAAI) calculated using the method proposed by Oser [17] with chicken egg protein as the reference standard, and the EAAIp (essential amino acid index—ideal amino acid profile of protein for porkers) based on the concept of ideal protein for pigs proposed by Muller in 1999, as cited by Grela [18].

### 2.2. Animals, Diets, and Experimental Procedures in Digestibility-Balance Trials

Digestibility-balance trials (experiments IA and IB) were performed using a simple balance method at the Animal Research Laboratory of the Department of Animal Nutrition and Feed Science, University of Warmia and Mazury in Olsztyn. A five-day experimental period proper was preceded by a seven-day pre-experimental (adaptation) period. The experimental materials comprised 20 male hybrid Danbred growing-finishing pigs divided into four groups of five animals for each experiment, IA (grower diets) and IB (finisher diets). The pigs were allocated to four groups using an analog method, according to the experimental design presented in Table 2. At the beginning of experiments IA (grower diets) and IB (finisher diets), the average body weight of pigs was 50 kg and 75 kg, respectively. The animals were housed in individual metabolism pens, with free access to water, and were fed crumbled feed that was offered wet (feed/water ratio of 1:1). The animals were fed complete grower (Table 3) and finisher diets (Table 3) in the grower and finisher phases of fattening, respectively. The composition of diets in digestibility-balance trials was identical to that in the feeding trial (experiment II). Daily feed allowance was determined based on the Feeding Guidelines for Pigs [15] at a target daily gain of 850 g.

During the five-day experimental period of digestibility-balance trials (experiments IA and IB), feces and urine were collected quantitatively. Two samples of 5% each were collected from feces produced during 24 h. One sample was dried, and the other sample was frozen. The content of dry matter, crude ash, crude fat, crude fiber, and gross energy was determined in the average dried fecal sample. Nitrogen content was determined in the average frozen fecal sample. Urine was preserved with 20% sulfuric acid to maintain pH below 2.0, and 5% samples were collected to determine N content.

The coefficients of apparent digestibility of dietary protein, fat, fiber, N-free extracts, and energy were calculated based on the chemical composition of feces and diets, and nutrient intake and output, with the use of the following equation: DC (%) = NI-NO/NI × 100; where: DC—digestibility coefficient (%), NI—nutrient intake (g), NO—fecal nutrient output (g), and NO/NI—nutrient digestibility (g).

Nitrogen retention was determined based on dietary N intake and fecal and urinary N excretion for each fattening phase and different vegetable protein sources of the diets. Nitrogen utilization was calculated based on the apparent biological value of protein expressed as N retention relative to N digested.

### 2.3. Animals, Diets, and Experimental Procedures in the Feeding Trial

A feeding trial (experiment II) was conducted at the Animal Research Laboratory of the Department of Animal Nutrition and Feed Science, University of Warmia and Mazury in Olsztyn. The experimental materials comprised 28 male hybrid Danbred growing-finishing pigs. The animals were housed in individual pens with a surface area of 2.6 m^2^ (length 1.7 m × width 0.95 m × height 1 m) with a slatted floor equipped with nipple drinkers. The pigs were allocated to four groups using an analog method, according to the experimental design presented in Table 2. The initial body weight of pigs was 26 kg, and their final body weight was 104 kg. Fattening was divided into two phases (grower: 26–67 kg BW and finisher: 67–104 kg BW). The animals were fed complete grower (Table 3) and finisher diets (Table 3) in the grower and finisher phases of fattening, respectively. Pigs were fed crumbled feed that was offered wet (feed/water ratio of 1:1), and had free access to water. The composition and nutritional value of grower and finisher diets are presented in Table 3. The nutritional value of diets was determined based on the Feeding Guidelines for Pigs [14].

Fattening performance was expressed as daily gains in the grower phase (26–67 kg BW), finisher phase (67–104 kg BW), and the entire fattening period (26–104 kg BW). The feed conversion ratio (FCR) was calculated based on feed, metabolizable energy, and digestible protein intake per kg of body weight gain in both fattening phases.

At the completion of the feeding trial, carcass quality was evaluated. All 28 pigs subjected to the feeding trial were slaughtered at a body weight of around 104 kg in the “Warmia” Meat Processing Plant in Biskupiec. Immediately after slaughter, carcass dressing percentage was determined, and carcass conformation and fat cover were evaluated based on the EUROP system criteria, with the use of a CGM 100 ultrasonic device equipped with an optical probe. Measurements were performed at the level of the last thoracic vertebra, 7 cm from the dorsal midline.

### 2.4. Selected Serum Biochemical Parameters

Blood was sampled from five pigs per group to determine indicators of selected biochemical processes in pigs fed diets with different vegetable protein sources. Blood samples for analyses were collected from the anterior vena cava of live animals before the morning feeding at the end of the experimental period proper in both digestibility-balance trials (experiments IA and IB). Whole blood samples were collected into test tubes containing a chemically neutral clotting activator. Blood samples were allowed to clot at room temperature for 30 min, and they were centrifuged in the MPW-348 centrifuge at 2000 rpm for 10 min. The serum samples for testing were stored at −20°C. Carbohydrate metabolism was analyzed based on serum glucose concentrations. Protein metabolism was analyzed based on serum total protein and ammonium N levels. Liver function was evaluated based on the activities of aspartate aminotransferase (AST), alanine aminotransferase (ALT), and gamma-glutamyltransferase (GGT). Kidney function was evaluated based on serum creatinine concentrations.

### 2.5. Laboratory Analyses

The content of major nutrients in feedstuffs, diets and fecal samples, and urinary N excretion were determined by standard methods [19].

Gross energy concentrations in feedstuffs, diets, and fecal samples were determined by adiabatic bomb calorimetry (IKA^®^ C2000 basic, Staufen, Germany). Metabolizable energy concentrations in diets were determined based on digestible nutrients content, using the Rostock Feed Evaluation System (RFES) formula [14].

The amino acid composition of protein and amino acid concentrations in the analyzed high-protein vegetable feed ingredients were determined with the use of an Amino Acid Analyzer AAA 400. The samples were hydrolyzed with 6M HCL at a temperature of 110 °C for 24 h. The concentrations of sulfur-containing amino acids were determined after sample oxidation with performic acid. Tryptophan content was determined in accordance with the Polish Standard [20].

The content of crude fiber, including neutral detergent fiber (NDF), acid detergent fiber (ADF), and acid detergent lignin (ADL) in feedstuffs, was determined by the method proposed by Van Soest et al. [21] using the FOSS TECATOR Fibertec 2010 System.

The concentrations of oligosaccharides in the seeds of YL cv. Taper and FB cv. Albus after extraction were determined by high-performance liquid chromatography (HPLC) using a refractive index RID-10A detector, Phenomenex Luna NH_2_ column, a mobile phase of 65% acetonitrile, a flow rate of 1.2 mL/min, and an injection volume of 20 μL.

The concentrations of glucosinolates in 00-RSM samples were determined in the Shimadzu HPLC system using a Varian MetaCarb 67H column.

The tannin content of FB seeds was determined by colorimetry [22], and the content of trypsin inhibitors in GM-SBM was determined by spectrophotometry [23].

Serum glucose was estimated by the glucose oxidase method, serum ammonium N levels were determined using the kinetic method, total serum protein was analyzed using the biuret method, serum creatinine concentrations were determined by the Jaffe method, the activities of AST, ALT, and GGT were evaluated using the kinetic method using an ACCENT-200 automatic biochemistry analyzer and commercial Cormay kits. The results are expressed in SI units [24].

### 2.6. Statistical Analysis

The results of digestibility-balance and feeding trials (arithmetic means) were processed statistically by one-way analysis of variance (ANOVA) using licensed STATISTICA 12.0 software. The significance of differences between group means was estimated by Duncan’s multiple range test at significance levels of *p* ≤ 0.05 and *p* ≤ 0.01 [25].

## 3. Results

The chemical composition of the analyzed high-protein vegetable feedstuffs is presented in Table 1. Similar values were reported by [12,26,27,28]. Protein from FB seeds cv. Albus had the highest lysine content (5.87 g/16 g N). Lysine concentrations were lower in protein from GM-SBM, 00-RSM, and YL seeds cv. Taper (5.20, 4.68, and 5.33 g/16 g N, respectively). Protein from legumes seeds had a low content of sulfur-containing amino acids (1.34 to 2.09 g/16 g N) and tryptophan (0.75 to 0.88 g/16 g N). The concentration of methionine+cystine was highest (4.28 g/16 g N) in 00-RSM.

The EAAI of GM-SBM and 00-RSM protein was 71.0 and 70.8, respectively. The values of the EAAI were considerably lower in legume seeds (YL—65.1 and FB—68.2). The nutritional value of protein determined based on the EAAIp was higher in 00-RSM and (89.7) and GM-SBM (87.3), and lower in low-tannin FB seeds (81.7) and low-alkaloid YL (75.1).

The data in Table 3 shows that complete grower and finisher diets were characterized by high nutritional value, consistent with Pig Nutrient Requirements [15]. Total protein concentrations ranged from 170 to 171.7 g/kg in grower diets and from 148 to 149.3 g/kg in finisher diets. The analyzed vegetable protein sources had a minor effect on the digestible protein content of diets due to differences in protein digestibility. Digestible protein concentrations were highest in grower controls (S-c) (150 g/kg) and R + YL (150 g/kg) diets and lowest in diet R (144 g/kg). A similar trend was noted in finisher diets, where digestible protein concentration was highest in diet S-c (129 g/kg), followed by diet R + YL (128 g/kg) and diet R (125 g/kg). The experimental factor had no significant effect on protein quality evaluated based on the concentrations of essential amino acids (lysine, methionine+cystine, threonine, and tryptophan). The tested vegetable protein sources induced minor differences in metabolizable energy concentrations, which were high in both grower (12.75 to 12.86 MJ/kg) and finisher (12.72 to 12.85 MJ/kg) diets.

Grower diets where 50% of GM-SBM protein was replaced with 00-RSM protein significantly (*p* ≤ 0.05) decreased the digestibility of total protein and energy (Table 4), which was determined at 87.8% vs. 84.5% and 86.4% vs. 84.9%, in diet S-c vs. diet R, respectively.

The replacement of GM-SBM protein with 00-RSM protein combined with protein from low-tannin FB seeds (diet R + FB) or low-alkaloid YL seeds (diet R + YL) had no significant negative effect on protein or energy digestibility compared to the S-c group. The obtained values of these indicators were high and amounted to 87.3% and 87.4% for protein, while for energy, they ranged from 86.3% to 86.8%.

Crude fiber digestibility was significantly (*p* ≤ 0.05) higher in diets R + FB and R + YL than in diet R (46.0% and 47.1% vs. 43.3%, respectively).

Daily N balance data for pigs fed grower diets is presented in Table 4. Fecal N excretion was significantly (*p* ≤ 0.05) higher in pigs fed diet R (11.0 g/day) than in the animals receiving the control diet S-c (8.6 g/day) and experimental diets R + FB and R + YL (9.0 g/day each). Urinary N excretion tended to be higher in the control group S-c and experimental groups R + FB and R + YL than in group R (27.5, 28.3, and 27.1 vs. 25.9 g/day, respectively). Fecal and urinary N losses had no negative effect on N retention or utilization. Nitrogen retention was high in all groups, ranging from 33.5 to 35.0 g/day (*p* = 0.104). Nitrogen retention as a percentage of N intake (47.2% to 49.2%, *p* = 0.182) and N digested (54.4% to 56.4%, *p* = 0.231) was also high in all groups. However, it should be stressed that N retention (35.0 g/day) and utilization (49.2% relative to N intake and 56.4% relative to N digested) were highest in growing pigs fed protein from 00-RSM and YL seeds (diet R + YL).

Table 5 present the digestibility of nutrients from finisher pig diets where 100% of GM-SBM protein was replaced with 00-RSM protein, alone or in combination with protein from FB and YL seeds. No significant differences in total protein digestibility were found between groups (*p* = 0.694), but lower values were noted in groups R (84.4%) and R + FB (84.9%) than in groups R + YL (86.0%), and S-c (86.8%). The experimental factor had no significant positive effect on the digestibility of crude fiber, crude fat, N-free extractives, or energy.

Daily N balance in pigs fed finisher diets is presented in Table 5. Fecal N excretion was highest in pigs fed diet R + FB (11.1 g/day), and this value was significantly (*p* ≤ 0.05) higher than in the animals receiving diets R + YL (9.7 g/day) and S-c (9.2 g/day). Urinary N excretion was significantly lower in pigs fed diets R (00-RSM), R + FB (00-RSM + FB seeds) and R + YL (00-RSM + YL seeds) than in those fed the control diet S-c, and the respective values were 24.7, 25.1 and 24.9 g/day vs. 26.2 g/day.

Vegetable protein sources in finisher diets had no significant effect on N retention (*p* = 0.180) in pigs, which was high in all groups, ranging from 34.6 g/day in group S-c to 33.3 g/day in group R + FB. The experimental factor exerted no significant (*p* = 0.529) influence on the apparent biological value of protein expressed as N retention relative to N digested in pigs fed finisher diets with vegetable protein sources alternative to GM-SBM, which ranged from 56.3% to 57.8%.

Selected serum biochemical parameters in pigs fed complete grower and finisher diets are presented in Table 4 and Table 5, respectively. Serum glucose concentrations were similar in all groups, ranging from 5.25 to 5.45 mmol/l (*p* = 0.121) in pigs fed grower diets and from 4.52 to 4.66 mmol/l (*p* = 0.432) in pigs fed finisher diets, which indicates that all diets had similar energy values. Total protein concentrations were highest in the control group S-c (63.78 and 67.52 g/l in grower and finisher diets, respectively) and lower in experimental groups R, R + FB, and R + YL, ranging from 61.58/65.68 to 62.72/67.18 g/l, but the noted differences were not statistically significant. Ammonium N levels tended to increase in the blood serum of pigs fed grower diets R + FB and R + YL (5.47 and 5.85 mmol/l, respectively) relative to those fed diet R (4.16 mmol/l). Similar trends were observed in pigs fed finisher diets. The above values point to lower utilization of protein from grower and finisher diets R + FB and R + YL. This fact was confirmed by N balance parameters (Table 4 and Table 5), showing that urinary N excretion was significantly higher in groups R + FB and R + YL (particularly in the grower phase) than in group R, characterized by the lowest urinary N excretion.

Serum creatinine concentrations in pigs fed grower and finisher diets remained within the lower limit of the normal range (96.58–102.90/123.07–135.55 μmol/l, respectively) [24]. The above values show that the experimental factor had no negative influence on kidney function in growing-finishing pigs.

The activities of AST, ALT, and GGT in the blood serum of pigs fed grower and finisher diets with different vegetable protein sources were similar in all groups. Serum AST levels remained within the reference ranges [24], whereas serum ALT and GGT levels slightly exceeded the upper limit of the normal ranges. The noted values indicate that the biochemical parameters of liver function were not adversely affected by the experimental factor.

The fattening performance of pigs fed complete grower and finisher diets where 50% or 100% of GM-SBM protein was replaced with 00-RSM protein, alone or in combination with protein from FB and YL seeds, is presented in Table 6.

At the beginning of the feeding trial, pigs had similar average initial body weights (26.21 kg to 26.57 kg). At the end of the grower phase of fattening, no significant (*p* = 0.509) differences in the average body weight of pigs were found between groups, and the noted values ranged from 65.54 kg in the control group S-c to 68.56 kg in group R + YL. The average final body weight of pigs did not differ significantly (*p* = 0.496) among groups.

Complete grower diet R + YL (00-RSM + YL seeds) significantly (*p* ≤ 0.05) increased the average daily gains of pigs in the first phase of fattening (26–67 kg BW), compared with the control diet S-c (GM-SBM) and diet R (00-RSM), which reached 981 g vs. 915 g and 929 g. In the second phase of fattening (67–104 kg BW), finisher diets R + FB and R + YL tended to improve the growth rate of pigs (*p* = 0.074), relative to diet R. An analysis of average daily gains over the entire fattening period (26–104 kg BW) revealed that pigs fed diet R + YL were characterized by a significantly higher growth rate (1027 g/day) than the animals fed diets R (970 g/day, difference of 5.9%) and R + FB (986 g/day, difference of 4.3%).

In the grower phase, FCR expressed as feed intake per kg of body weight gain was highly significantly (*p* ≤ 0.05) lower (2.46 kg/kg) in group R + YL than in the remaining groups (group S-c—2.68 kg/kg, group R—2.61 kg/kg, group R + FB—2.60 kg/kg). In the finisher phase, feed intake per kg of body weight gain was significantly (*p* ≤ 0.05) higher in groups R and R + FB than in groups S-c and R + YL. During the entire fattening period, the FCR was significantly (*p* ≤ 0.05) affected by the evaluated vegetable protein sources. The FCR ratio in group R + YL was 7.5% and 5.5% lower than in groups R and R + FB, respectively, whereas the difference relative to group S-c was not statistically significant.

An analysis of metabolizable energy intake per kg of body weight gain revealed significant (*p* ≤ 0.05) differences between groups in the grower phase of fattening. The lowest value—31.36 MJ/kg was noted in pigs fed diet R + YL, compared with 33.56, 34.27, and 33.30 MJ/kg in groups S-c, R, and R + FB, respectively. In the finisher phase, metabolizable energy intake per kg of body weight gain tended (*p* ≤ 0.05) to be higher in groups R (39.17 MJ/kg) and R + FB (39.19 MJ/kg) than in group S-c (36.63 MJ/kg). During the entire fattening period, a significant tendency (*p* ≤ 0.05) to better utilization of metabolizable energy was noted in diet R + YL, compared with diets R and R + FB. The difference between diets R + YL and S-c was not statistically significant.

An analysis of digestible protein utilization per kg of body weight gain revealed differences between groups. In the grower phase, the value noted in group R + YL (369 g/kg) was significantly (*p* ≤ 0.05) lower than those observed in groups S-c, R, and R + FB (392, 386, and 387 g/kg, respectively). In the finisher phase, digestible protein utilization tended (*p* = 0.081) to be worse in groups R, R + FB, and R + YL vs. group S-c. During the entire fattening period, digestible protein intake per kg of body weight gain was lowest (*p* = 0.073) in group R + YL (359 g/kg), compared with groups S-c (372 g/kg), R (375 g/kg) and R + FB (377 g/kg), and the noted differences reached 3.6%, 4.0%, and 5.0%, respectively.

Carcass quality characteristics are presented in Table 6. No significant (*p* = 0.620) differences in dressing percentage were found between pigs fed diets with different vegetable protein sources. The average dressing percentage was lowest in the control group S-c (73.99%) and highest in group R + FB (75.19%). Pigs fed the R + FB diet were characterized by the highest average backfat thickness (17.00 mm). In the remaining groups, the values of this parameter ranged from 16.14 mm to 16.39 mm (*p* = 0.461). Carcass lean content ranged from 56.16% in group R to 56.74% in group R + YL (*p* = 0.287). Meat from pigs receiving diet R + FB was characterized by the highest value of loin eye height in the longissimus dorsi muscle (57.43 mm), whereas meat from pigs receiving diet R was characterized by the lowest value of this parameter (54.49 mm). The differences between group means were not statistically significant (*p* = 0.312).

## 4. Discussion

Previous research into the partial replacement of GM-SBM with 00-RSM, FB seeds or YL seeds in grower pig diets revealed differences in nutrient digestibility. In the present study, 00-RSM protein combined with protein from FB seeds or YL seeds as a substitute for 50% of GM-SBM protein in diets for growing pigs had a beneficial influence on nutrient digestibility. In the analyzed diets, the inclusion levels of 00-RSM, FB seeds, and YL seeds were 6%, 10%, and 6%, respectively. Such a combination contributed to a significant increase in the digestibility of protein and energy from grower diets, compared with the diet containing 12% of 00-RSM. It appears that the combined high-protein feedstuffs improved the quality of dietary protein due to the complementary effects exerted by nutritionally important, essential amino acids (methionine + cystine and lysine).

Similar to the current study, Hanczakowska and Świątkiewicz [29] reported no significant differences in total protein digestibility between the control SBM-based diet and diets containing YL and FB seeds. Jezierny et al. [30] demonstrated that the standardized ileal digestibility of total protein from SBM and YL seeds did not differ significantly. However, the digestibility of FB protein was highly significantly lower than the digestibility of SBM protein (76% vs. 87%). Purwin and Stanek [31] analyzed the nutrient digestibility of legume seeds and found that total protein digestibility was lower in a diet containing 33% of FB seeds than in the control diet (67.2% vs. 75.5%). The inclusion of YL seeds at 18% improved protein digestibility from 73.8% to 74.4%. In a study by Mariscal-Landín et al. [32], the apparent ileal digestibility of total protein, including most amino acids, decreased significantly when RSM was added to the diet. Eklund et al. [33] and Torres-Pitarcha et al. [27] demonstrated that 00-RSM led to a decrease in total protein digestibility. The lower protein digestibility of RSM, compared with SBM, may be related to the high content of fiber, including lignin, in the seed coat of rapeseeds because amino acids absorbed by crude fiber are less available in the small intestine [27].

In the present study, the coefficient of crude fiber digestibility was higher in groups R + FB and R + YL than in groups S-c and R. According to Hanczakowska and Świątkiewicz [29], the inclusion of FB and YL seeds in pig diets significantly increases crude fiber digestibility, from 29.1% to 34.5% and 39.2%, respectively. In a study of growing-finishing pigs fed legume-based diets, Stanek et al. [34] found that lower YL content (11.5%) increased, but higher YL content (18%) decreased crude fiber digestibility. Faba bean seeds led to a significant decrease in energy digestibility, whereas YL seeds had no significant effect on energy digestibility. Torres-Pitarch et al. [27] demonstrated that 00-RSM decreased the crude fiber digestibility of fattening pig diets.

An analysis of the nutrient digestibility of finisher pig diets revealed a non-significant positive effect of 00-RSM combined with YL seeds on the digestibility of protein, fat, and energy, compared with 00-RSM applied alone. Hanczakowska and Świątkiewicz [29] found no differences in the digestibility of total protein, fat, or crude fiber between pig diets containing SBM, FB, or YL seeds. However, the digestibility coefficient of N-free extracts was highly significantly higher in the group fed FB seeds compared with the remaining groups. Purwin and Stanek [34] reported that FB seeds used as a complete substitute for SBM in finisher pig diets at 35% increased crude fat digestibility (59.0% vs. 62.0%) but decreased energy digestibility (82.2% vs. 79.7%). Crude fiber digestibility was improved (from 37.6% to 46.8%) by replacing SBM with YL seeds at 18%.

In the present study, partial (50% in grower diets) and total (100% in finisher diets) replacement of GM-SBM protein with protein from 00-RSM, FB, and YL seeds had no adverse effect on the indicators of carbohydrate and protein metabolism or the biochemical parameters of liver and kidney function. However, 00-RSM used as a partial (50%) or total (100%) substitute for GM-SBM had a negative influence on fattening performance. Complete grower and finisher diets contained 12% and 13% of 00-RSM, respectively. Nevertheless, the average daily gain for the entire fattening period was high, at 970 g/day, and feed intake per kg body weight gain was 2.79 kg. The values of the above parameters were lower than in the control group fed SBM-based diets, but the noted differences were not statistically significant. The present results corroborate the findings of Sobotka et al. [4], McDonnell et al. [35], and Xie et al. [36]. According to McDonnell et al. [35], the results of previously published studies investigating 00-RSM as a substitute for SBM in diets for growing-finishing pigs are inconclusive. Some studies have demonstrated that 00-RSM can be included in pig diets at up to 20% without compromising growth performance, whereas others have shown that RSM inclusion levels of 10% to 20% may adversely affect fattening efficiency. These contradictory findings could result from differences in glucosinolate concentrations in rapeseeds. According to Torres-Pitarch et al. [27], a high nutrient content in pig diets is not always associated with better performance. The nutritional value of RSM may vary depending on processing technology, including temperature during toasting [36]. The negative impact of RSM on the growth performance of pigs, reported by Torres-Pitarch et al. [27], could result from its high glucosinolate content (15.58 μmol/g), which was more than two-fold higher than that determined in the present study.

In this experiment, protein from 00-RSM combined with low-tannin FB seeds had a non-significant positive effect on the analyzed parameters. Grower and finisher diets containing 10% and 12% FB seeds, respectively, combined with 6% 00-RSM, had no significant negative influence on fattening performance compared with SMB-based diets. A combination of FB seeds and 00-RSM increased the growth rate of pigs by 4.5%, particularly in the grower phase when the average daily gain reached 956 g in group R + FB, compared with 915 g in group R. In the finisher phase, the average daily gain was similar in both groups, at 994 g and 986 g, respectively. The present results are consistent with the findings of other authors [29,37,38,39] who evaluated the effects exerted by the above dietary protein sources when applied alone.

In the current study, partial (50%) and total (100%) replacement of GM-SBM protein with protein from 00-RSM and YL seeds had a significant positive influence on fattening performance. In grower and finisher diets, 6% of 00-RSM was combined with 6% and 7% of YL seeds, respectively. In group R + YL, the average daily gain was high in the grower (981 g) and finisher (1030 g) phases and over the entire fattening period (1027 g). Similar to the growth rate of pigs, FCR was also significantly better in group R + YL than in groups R and R + FB.

Roth-Maier et al. [40] evaluated the effect of grower diets containing 20% of sweet lupine as a substitute for SMB on the growth performance and carcass characteristics of growing-finishing pigs and reported higher daily gains and body weights and a better FCR in the growing period in the experimental group, which corresponds to our findings. In the work of Hanczakowska and Świątkiewicz [29], YL seeds included at 8% and 12% in grower and finisher diets as a complete substitute for SBM had no significant positive effect on fattening performance. Similar observations were made by other authors [41,42], who found that YL could be used as a sole protein source in pig diets without compromising growth performance.

In the current study, alternative vegetable protein sources partially and totally replacing SBM in grower and finisher diets had no significant effect on carcass quality traits. Similar results were reported by other authors who investigated 00-RSM [4,34,43], YL seeds [30,42], and FB seeds [39,40,41]. However, Degola [44] found that fat deposition increased significantly in the carcasses of pigs fed diets containing 20% of FB seeds.

## 5. Conclusions

The results of this study indicate that partial (50% in grower diets) and total (100% in finisher diets) replacement of GM-SBM protein with 00-RSM protein (6%) combined with protein from FB seeds or YL seeds in pig diets has no adverse effect on nutrient and energy digestibility, nitrogen balance, serum carbohydrate and protein metabolism or the biochemical parameters of liver and kidney function. Protein from 00-RSM (6%) and FB seeds (10/12%) contributed to high daily gains and high feed conversion efficiency. Protein from 00-RSM (6%) and YL seeds (6/7%) in grower/finisher diets led to a further improvement in fattening performance. The analyzed vegetable protein sources had no negative influence on carcass quality. It can be concluded that 00-RSM protein combined with protein from low-tannin FB or low-alkaloid YL seeds can be valuable high-protein feed ingredients alternative to GM-SBM in pig nutrition.

## Figures and Tables

**Table 1 animals-11-00960-t001:** Chemical and amino acid composition of protein and the content of antinutritional factors in high-protein feed ingredients.

Specification	Soybean Meal ^1^ (GM-SBM)	00-Rapeseed Meal (RSM)	Faba Bean cv. Albus (FB)	Yellow Lupine cv. Taper (YL)
Chemical composition (% DM) ^2^ and energy value (MJ/kg) ^2^				
Dry matter	89.28	88.13	87.63	86.57
Crude protein	51.83	39.74	29.85	40.31
Crude fat	0.87	1.00	0.73	4.97
N-free extracts	30.61	33.61	53.16	26.33
Crude fiber	4.66	11.00	8.91	18.02
Neutral detergent fiber (NDF)	12.72	30.86	19.41	28.14
Acid detergent fiber (ADF)	8.00	22.32	13.50	2.25
Acid detergent lignin (ADL)	0.44	8.35	1.06	1.05
Hemicelluloses ^3^	4.73	8.54	5.91	25.89
Cellulose ^4^	7.56	13.97	12.44	1.19
Gross energy, MJ/kg	17.46	17.18	16.20	16.94
Energy digestibility (DCe),^5^ %	86.9	70.6	80.9	73.1
Content of antinutritional factors				
Trypsin inhibitors, mg/g DM ^2^	1.57	-	-	-
Glucosinolates, μmol/g dry-non fat mass ^2^	-	6.85	-	-
Tannins, g/kg DM ^2^	-	-	2.69	-
Alkaloids, g/kg DM ^2^	-	-	-	0.07
Oligosaccharides, mg/g DM ^2^	-	-	66.00	162.41
Amino acids (g/100 g of crude protein)				
Thr	3.18	3.94	3.51	2.65
Val	4.11	4.14	3.92	2.76
Cys	1.34	1.76	0.67	1.34
Met	1.78	2.52	0.66	0.75
Ile	3.87	4.11	3.51	3.38
Leu	6.89	6.12	6.88	7.50
Tyr	2.96	2.79	2.96	2.22
Phe	4.22	3.27	3.67	3.23
His	2.16	2.48	2.41	2.25
Lys	5.20	4.68	5.87	5.33
Arg	5.97	4.67	7.71	8.45
Trp	1.22	1.18	0.88	0.75
Chemical analysis of protein quality				
EAAI ^6^	71.0	70.8	68.2	65.1
CS ^7^	44.3 (Met+Cys)	59.5 (Ile)	20.5 (Met+Cys)	32.5 (Met+Cys)
EAAIp ^8^	87.3	89.7	81.7	75.1

^1^—genetically modified soybean meal; ^2^—mean values from the chemical analyses of feed ingredients; ^3^—NDF–ADF; ^4^—ADF-ADL; ^5^—0.984–0.00090 * NDF [16]; ^6^—essential amino acid index; ^7^—chemical score; ^8^—essential amino acid index—ideal amino acid profile of protein for porkers; - not determined.

**Table 2 animals-11-00960-t002:** Feeding trial design.

Group	Number of Animals	Source of Vegetable Protein ^1^
Grower diets —50% of protein from genetically modified soybean meal (GM-SBM) was replaced with protein from 00-rapeseed meal (00–RSM), faba bean seeds (FB), and yellow lupine seeds (YL)		
S-c	7	GM-SBM
R	7	GM-SBM + 00-RSM
R + FB	7	GM-SBM + 00-RSM + seeds of FB cv. Albus
R + YL	7	GM-SBM + 00-RSM + seeds of YL cv. Taper
Finisher diets—100% of protein from GM-SBM was replaced with protein from 00-RSM in 50%, and with FB and YL s in 50%		
S-c	7	GM-SBM
R	7	00-RSM
R + FB	7	00-RSM + seeds of FB cv. Albus
R + YL	7	00-RSM + seeds of YL cv. Taper

^1^—grower diets; S-c—control group–genetically modified soybean meal (GM-SBM); R—genetically modified soybean meal (GM-SBM) + 00-rapeseed meal (00-RSM); R + FB—genetically modified soybean meal (GM-SBM) + 00-rapeseed meal (00-RSM) + seeds of faba bean (FB) cv. Albus; R + YL—genetically modified soybean meal (GM-SBM) + 00-rapeseed meal (00-RSM) + seeds of yellow lupine (YL) cv. Taper; ^1^—finisher diets S-c—control group–genetically modified soybean meal (GM-SBM); R—00-rapeseed meal (00-RSM); R + FB—00-rapeseed meal (00-RSM) + seeds of faba bean (FB) cv. Albus; R + YL—00-rapeseed meal (00-RSM) + seeds of yellow lupine (YL) cv. Taper.

**Table 3 animals-11-00960-t003:** Composition and nutritional value of complete diets for growing-finishing pigs *(Experiments IA, IB, and II).*

Feed Ingredients	Experimental Diets							
	Grower (26–67 kg BW)				Finisher (67–104 kg BW)			
	S-c	R	R + FB	R + YL	S-c	R	R + FB	R + YL
Substitution of GM-SBM protein (%):	0.0	50.0	50.0	50.0	0.0	100	100	100
Wheat	40.00	38.00	36.00	38.00	44.0	41.00	39.00	42.00
Barley	40.30	37.47	35.93	37.96	43.60	41.57	39.12	41.25
GM-SBM	16.00	8.00	8.00	8.00	9.00	-	-	-
00-RSM	-	12.00	6.00	6.00	-	13.00	6.00	6.00
Faba bean cv. Albus	-	-	10.00	-	-	-	12.00	-
Yellow lupine cv. Taper				6.00	-	-	-	7.00
Rapeseed oil	1.00	1.80	1.40	1.30	1.00	2.00	1.50	1.30
L-lysine HCL (78%)	0.20	0.23	0.17	0.24	0.20	0.23	0.18	0.25
Vitamins+trace minerals ^1^	2.50	2.50	2.50	2.50	2.20	2.20	2.20	2.20
Nutritional value of diets (g/kg):								
ME ^2^ (MJ/kg)	12.86	12.79	12.81	12.75	12.81	12.72	12.85	12.77
Total protein	170.0	170.8	171.5	171.7	148.3	148.0	149.3	148.7
Digestible protein	150	144	149	150	129	125	127	128
Lysine	9.71	9.69	9.77	9.70	8.11	8.06	8.17	8.03
Methionine + cystine	6.01	6.21	6.05	6.17	5.41	5.67	5.43	5.50
Threonine	6.11	6.31	6.20	6.21	5.19	5.39	5.20	5.24
Tryptophan	2.21	2.43	2.20	2.27	1.90	2.10	1.98	1.92
Crude fiber	48.3	56.5	55.5	57.7	48.2	55.7	56.2	58.3
Calcium	7.51	7.57	7.40	7.52	6.30	6.67	6.50	6.63
Total phosphorus	5.10	5.61	5.43	5.44	5.10	5.43	5.19	5.21
Sodium	1.50	1.50	1.50	1.50	1.30	1.30	1.30	1.30

^1^—Provided per kilogram of diet: (L-Lysine 110 g; DL-Methionine 15 g; L-Threonine 16.7 g; Tryptophan 5.4 g; Valin 7.2 g; Ca 192 g; mineral and free P 50 g; Na 51 g; Mg 16 g; vit.: A 400,000 IU, D_3_ 80,000 IU, E 2200 mg, K 120 mg, B_1_ 80 mg, B_2_ 240 mg, B_6_ 120 mg, B_12_ 1200 μg, biotin 6000 μg. niacin 960 mg; pantothenic acid 480 mg; folic acid 40 mg; betaine 2480 mg; choline chloride 5360 mg; Fe 4000 mg; Zn 4000 mg; Mn 2400 mg; Cu 600 mg; I 32 mg; Se 12 mg; phytase);^2^—metabolizable energy.

**Table 4 animals-11-00960-t004:** Digestibility of nutrients and energy in complete grower diets, nitrogen balance, and serum biochemical parameters of growing pigs ((Experiment I A).

Specification		Grower Diets ^1^					
		S-c	R	R + FB	R + YL	SEM	*p*-Value
Substitution of GM-SBM Protein (%):		0.0	50	50	50		
Number of Pigs in the Experimental Diets (n)		5	5	5	5		
Digestibility coefficients (%):							
Total protein		87.8 ^a^	84.5 ^b^	87.3 ^a^	87.4 ^a^	0.499	0.039
Total fat		75.4	76.8	74.8	77.1	1.859	0.086
Total crude fiber		45.8 ^a^	43.3 ^b^	46.0 ^a^	47.1 ^a^	1.440	0.044
N-free extracts		91.7	90.9	91.7	91.6	0.173	0.173
Gross energy		86.4 ^a^	84.9 ^b^	86.8 ^a^	86.3 ^a^	0.320	0.041
Daily nitrogen balance (g/day):							
Intake		70.6	70.9	71.0	71.1	0.043	0.230
Fecal excretion		8.6 ^b^	11.5 ^a^	9.0 ^b^	9.0 ^b^	1.742	0.036
Digestion		62.0 ^a^	59.4 ^b^	62.0 ^a^	62.1 ^a^	0.892	0.027
Urinary excretion		27.5 ^a^	25.9 ^b^	28.3 ^a^	27.1 ^a^	3.303	0.026
Retention		34.5	33.5	33.7	35.0	0.634	0.104
Nitrogen retention relative to (%):							
N intake		48.9	47.2	47.5	49.2	0.882	0.182
N digested		55.6	56.3	54.4	56.4	1.033	0.231
	Reference values ^3^	Serum biochemical parameters ^2^					
Glucose (mmol/l)	2.5–5.6	5.45	5.44	5.45	5.25	0.111	0.121
Total protein (g/l)	59–74	63.78	61.58	60.48	62.72	0.603	0.511
Ammonium nitrogen (mmol/l)	3.3–6.6	5.28	4.16	5.47	5.85	0.198	0.168
Creatinine (μmol/l)	88.4–238.	99.58	101.58	96.58	102.90	1.239	0.439
AST ^4^ (U/l)	16–65	33.20	36.00	31.80	39.20	1.262	0.352
ALT ^5^ (U/l)	9–43	49.80	51.60	46.80	51.60	1.488	0.658
GGT ^6^ (U/l)	16–30	40.20	39.00	32.00	38.60	1.242	0.322

^1^ S-c—control group–genetically modified soybean meal (GM-SBM); R—genetically modified soybean meal (GM-SBM) + 00-rapeseed meal (00-RSM); R + FB—genetically modified soybean meal (GM-SBM) + 00-rapeseed meal (00-RSM) + seeds of faba bean (FB); R + YL—genetically modified soybean meal (GM-SBM) + 00-rapeseed meal (00-RSM) + seeds of yellow lupine (YL);^2^—blood was sampled from pigs at the end of digestibility-balance trials; ^3^—reference values of blood biochemical parameters [24]); ^4^—aspartate aminotransferase; ^5^—alanine aminotransferase; ^6^—γ-glutamyltransferase; SEM—standard error of mean; ^a. b^—within rows; values with different letters are significantly different (*p* ≤ 0.05).

**Table 5 animals-11-00960-t005:** Digestibility of nutrients and energy in complete finisher diets, nitrogen balance, and serum biochemical parameters of finishing pigs (Experiment IB).

Specification			Finisher Diets ^1^					
			S-c	R	R + FB	R + YL	SEM	*p*-Value
Substitution of GM-SBM Protein (%):			0.0	100	100	100		
Number of Pigs in the Experimental Diets			5	5	5	5		
	Digestibility coefficients (%):							
Total protein			86.8	84.4	84.9	86.0 0.	394	0.694
Total fat			84.6	82.2	86.0	87.4	1.202	0.420
Total crude fiber			42.2	40.9	39,6	41.4	1.365	0.236
N-free extracts			90.5	90.6	90.7	90.3	0.215	0.318
Gross energy			84.7	83.9	84.6	84.3	0.295	0.195
	Daily nitrogen balance (g/day):							
Intake			70.0	68.8	69.5	68.8	0.128	0.642
Fecal excretion			9.2 ^b^	10.4 ^b^	11.1 ^a^	9.7 ^b^	0.264	0.026
Digestion			60.8	58.4	58.4	59.1	0.192	0.526
Urinary excretion			26.2 ^b^	24.7 ^a^	25.1 ^a^	24.9 ^a^	1.064	0.044
Retention			34.6	33.7	33.3	34.2	1.080	0.180
	Nitrogen retention relative to (%):							
N intake			49.4	48.9	47.3	49.7	1.575	0.635
N digested			56.9	57.7	56.3	57.8	1.825	0.529
		Reference values ^3^	Serum biochemical parameters ^2^					
Glucose		2.5–5.6	4.61	4.66	4.52	4.57	0.084	0.432
Total protein (g/l)		59–74	67.52	65.68	67.08	67.80	0.359	0.659
Ammonium nitrogen (mmol/l)		3.3–6.6	5.49 ^Ac^	3.85 ^Bb^	4.21 ^Ba^	5.09 ^Ad^	0.168	0.010
Creatinine (μmol/l)		88.4–238.7	130.19	128.07	123.07	135.55	1.654	0.854
AST ^4^ (U/l)		16–65	39.00	37.40	33.40	42.80	1.212	0.412
ALT ^5^ (U/l)		9.0–43	61.40	50.80	49.20	55.00	1.702	0.570
GGT ^6^ (U/l)		16–30	52.20	46.60	54.80	54.40	2.130	0.613

^1^—S-c—control group–genetically modified soybean meal (GM-SBM); R—00-rapeseed meal (00-RSM); R + FB—00-rapeseed meal (00-RSM) + seeds of faba bean (FB); R + YL—00-rapeseed meal (00-RSM) + seeds of yellow lupine (YL); ^2^—blood was sampled from pigs at the end of digestibility-balance trials; ^3^—reference values of blood biochemical parameters [25]; ^4^—aspartate aminotransferase; ^5^—alanine aminotransferase; ^6^—γ-glutamyltransferase; SEM—standard error of mean; ^a. b.^—within rows, values with different letters are significantly different (*p* ≤ 0.05); ^A. B.^—within rows, values with different letters are significantly different (*p* ≤ 0.01).

**Table 6 animals-11-00960-t006:** Fattening performance and carcass quality traits in growing-finishing pigs (Experiment II).

Specification	Group ^1^					
	S-c	R	R + FB	R + YL	SEM	*p*-Value
Substitution of GM-SBM Protein in Grower/Finisher Diets (%):	0/0	50/100	50/100	50/100		
Number of pigs in the experimental group (n)	7	7	7	7		
Initial body weight (kg)	26.29	26.21	26.57	26.36	1.870	0.870
Intermediate body weight (kg)	66.11	65.54	67.69	68.56	0.909	0.509
Final body weight (kg)	105.11	103.21	104.78	104.39	1.269	0.496
Body weight of pigs	Average daily gain (g/day):					
26–67 (kg)	929 ^b^	915 ^b^	956	981 ^a^	7.412	0.041
67–104 (kg)	1055	986	994	1030	9.646	0.074
26–104 (kg)	1001	970 ^b^	986 ^b^	1027 ^a^	6.181	0.044
Body weight of pigs	Feed conversion ratio (kg/kg):					
26–67 (kg)	2.61 ^a^	2.68 ^a^	2.60 ^a^	2.46 ^b^	0.011	0.018
67–104 (kg)	2.86 ^b^	3.08 ^a^	3.05 ^a^	2.98	0.021	0.029
26–104 (kg)	2.67	2.79 ^a^	2.73 ^a^	2.58 ^b^	0.016	0.022
Body weight of pigs	Metabolizable energy utilization (MJ/kg):					
26–67 (kg)	33.56 ^a^	34.27 ^a^	33.30 ^a^	31.36 ^b^	0.203	0.033
67–104 (kg)	36.63 ^b^	39.17 ^a^	39.19 ^a^	38.05	0.367	0.049
26–104 (kg)	34.27	35.58 ^a^	35.02 ^a^	32.92 ^b^	0.263	0.036
Body weight of pigs		Digestible protein utilization (g/kg):				
26–67 (kg)	392 ^a^	386 ^a^	387 ^a^	369 ^b^	2.575	0.042
67–104 (kg)	369	385	387	381	4.124	0.081
26–104 (kg)	372	375	377	359	3.163	0.073
Carcass quality traits						
Dressing percentage (%)	73.99	73.85	75.19	74.73	0.320	0.620
Backfat thickness (mm)	16.29	16.39	17.00	16.14	0.264	0.461
Lean content (%)	56.66	56.16	56.26	56.74	0.325	0.287
Loin eye height in the uscle longissimus dorsi (mm)	55.00	54.49	57.43	56.29	1.309	0.312

^1^—S-c—control group–genetically modified soybean meal (GM-SBM); R—genetically modified soybean meal (GM-SBM) + 00-rapeseed meal (00-RSM)/00-rapeseed meal (00-RSM); R + FB—genetically modified soybean meal (GM-SBM) + 00-rapeseed meal (00-RSM) + seeds of faba bean (FB)/00-rapeseed meal (00-RSM) + seeds of faba bean (FB); R + YL—genetically modified soybean meal (GM-SBM) + 00-rapeseed meal (00-RSM) + seeds of yellow lupine (YL)/00-rapeseed meal (00-RSM) + seeds of yellow lupine (YL); SEM—standard error of mean; ^a. b^—within rows, values with different letters are significantly different (*p* ≤ 0.05).

## Data Availability

Data confirming the results are provided by the authors of the paper. But I exclude this entry.

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
