# Peer review of "The Effect of Replacing Genetically Modified Soybean Meal with 00-Rapeseed Meal, Faba Bean and Yellow Lupine in Grower-Finisher Diets on Nutrient Digestibility, Nitrogen Retention, Selected Blood Biochemical Parameters and Fattening Performance of Pigs"

_animals, 2021, doi:10.3390/ani11040960_

Round 1

Reviewer 1 Report

Comment 1 - relates to Introduction

In the Introduction, the authors write about BSE and refer to the position of literature no. 1. The information is too extensive and partially inconsistent with the research. The withdrawal of animal meals years ago as a feed material for pig mixes was one of the factors that prompted the search for replacement protein components for mixes. The choice fell on legumes. Currently, the essence of the work is to test them as substitutes for post-extraction soybean meal in pig fattening. I propose to shorten (edit) the following paragraph.

Comment 2 - relates to Materials and Methods 

Information on the study of blood biochemical indicators is incomplete. How was blood collected and how was the material (serum) prepared for analysis? Has it been stored (frozen)? Or maybe the analysis was performer immediatelly after collection? Please specify the conditions of blood centrifugation and serum storage for biochemical analyzes.

Comment 3 - relates to Discussion & Materials and Methods

The key words include: Carcass quality. There are few results relating to the quality of the carcass (Table 6). In the discussion, the authors referred to several works (items 4, 30, 39) in which fatteners were slaughtered with body weight of 113-115 kg. Question - what were the authors of the assessed work guided by when slaughtering fatteners at very low body weight - approx. 104 kg? Currently, the slaughtering of fatteners is recommended (prefered) and often performed at body weight of  120-130 kg.

Author Response

Response to Reviewer 1

Comment 1 - relates to Introduction

As suggested by the Reviewer, the following paragraph has been removed from the Introduction section: Pigs have high protein requirements, therefore cereal-based diets must also contain high-protein components such as meals (including meat and bone meals) and legume seeds. Due to the health risk posed by bovine spongiform encephalopathy (BSE) and transmissible spongiform encephalopathies (TSE), a series of measures have recently been adopted in Europe to prevent and control the spread of these diseases. A total ban on feeding meat and bone meals to farm animals was introduced in the European Union (EU) in 2001 [1]. The ban had a negative effect on the supply of high-protein animal feeds in the EU member states, including Poland. Therefore, genetically modified soybean meal (GM-SBM) was imported to make up for the negative trade balance on high-protein ingredients used in monogastric nutrition [2-3].

and it has been replaced with the following paragraph:

Pigs have high protein requirements, therefore cereal-based diets must also contain high-protein components such as meals, including imported genetically modified soybean meal (GM-SBM) [1-2]. Recent years have witnessed considerable breeding progress in the genetic improvement of legumes. The seeds of new legume varieties are characterized by higher and more stable yields, and a lower content of antinutritional factors [3]. Therefore, they can be fed to pigs to make up for the negative trade balance on high-protein ingredients used in monogastric nutrition.

Comment 2 - relates to Materials and Methods, subsection 2.4. Selected serum biochemical parameters

The following information has been provided:

Whole blood samples were collected into test tubes containing chemically neutral clotting activator. Blood samples were allowed to clot at room temperature for 30 min, and they were centrifuged in the MPW-348 centrifuge at 2000 rpm for 10 min. The serum samples for testing were stored at -20°C.

Comment 3 - relates to Discussion & Materials and Methods - explanation

We agree that the slaughtering od fatteners is recommended/preferred when they achieve higher body weight than those in our experiment where the animals had to be slaughtered earlier than intended due to unforeseen circumstances (sudden illness of one of the employees). However, an analysis of the effects exerted by the experimental factors (vegetable dietary protein sources) provided valuable information on the parameters of carcass quality in pigs slaughtered at lower than recommended body weight.

Reviewer 2 Report

Very complete work although somewhat difficult to understand. Congratulations on your work, below I make some minor annotations that make, above all, reference to the form of presentation of the results. I think that if the tables were improved, it would improve the work enormously.

The statistical design needs to be better explained. Why were two separate analyzes performed? What was changed was the p_value? Or the level of significance. Is not the same. On the other hand, nothing is expressed about the factors that have been taken into account when practicing anova. Was normality tests performed? Complete development of the statistical design is lacking.
Please explain correctly the number of individuals per experiment, treatment, etc. Sometimes it is not clear if we are talking about animals by group or by treatment. They should put the number of individuals next to each data. Without it and without the standard error, correct interpretations of the results cannot be made, nor can I correct the work correctly.
Is the format of the L behind the species correct?
Table 1: Typically less marked spacing is used between columns. The tables have to be self-explanatory. EAAO? CS? EAAIp? All the information must appear, in a self-explanatory way.
Table 2: It has a better format (repeat in Tab 1).
L 148, number in bold.
Table 4: Tables should be self-explanatory. It should not refer to another table. Everything must be explored S-C, R….
A dot appears above "substitution of ..."
Table 5: A dot appears in front of “specification” Tables must be self-explanatory. It should not refer to another table. Also, the superscript 1 does not appear.
Table 6: A dot appears in front of “specification” Tables must be self-explanatory. It should not refer to another table.
Sometimes they end the information in the table with points and sometimes they don't, please unify criteria.
In general, they must unify all the tables, format, definition, etc.
Overall: Why are there no standard errors? They should be given in order to compare treatments.
They do an ingestion and retention test with 28 animals. They do not seem few individuals to detect differences for example, in the ingestion.

Author Response

Response to Reviewer 2

We would like to explain that two separate one-way analyses of variance were performed to process statistically the results of:

 - two digestibility-balance trials aimed to determine the effects of experimental factors (vegetable protein sources and levels of GM-SMB protein substitution) on nutrient digestibility and nitrogen balance parameters in pigs fed grower and finisher diets.

- one feeding trial aimed to determine the effects of experimental factors on the fattening performance of pigs and carcass quality.

Due to the analyzed traits and parameters and the type of the conducted trials, two separate one-way analyses of variance had to be performed.

The test of the normality of the Shapiro-Wilka variable distribution was performed before starting the statistical analysis of the results of the analyzed traits.

The following data have been added to Tables 4, 5 and 6:

 - the number of animals in groups (n)

 - SEM values.

The abbreviations EAAI, CS and EAAIp have been explained in a footnote under Table 1.

The symbols of diets/groups have been explained in footnotes under Tables 4, 5 and 6.

All other comments made by the Reviewer have been takin into account, and the Tables have been modified accordingly.

Round 2

Reviewer 2 Report

the authors have modified the initial version.

Author Response

Bardzo dziękuję za dotychczasowe uwagi i sugestie, które przyczynił się do merytorycznego ulepszenia artykułu naukowego. Kolejne nie trafiły przez Recenzentę. Odmówię pragnę po prawej stronie, że artykuł został poprawiony o kolejną poprawkę redakcyjną Recenzję przez redaktora naukowego czasopisma.
